# Feasibility of using low-cost markerless motion capture for assessing functional outcomes after lower extremity musculoskeletal cancer surgery

**Sherron Furtado**[1,2]*, **Brook Galna**[3,4], **Alan Godfrey**[5], **Lynn Rochester**[4,6], **Craig Gerrand**[7]

**1** Department of Orthopaedics and Musculoskeletal Science, University College London, London, United Kingdom, **2** Therapies and Department of Orthopaedic Oncology, London Sarcoma Service, Royal National Orthopaedic Hospital NHS Trust, Stanmore, United Kingdom, **3** School of Allied Health (Exercise Science), Murdoch University, Perth, Australia, **4** Translational and Clinical Research Institute, Newcastle University, Newcastle upon Tyne, United Kingdom, **5** Computer and Information Science Department, Northumbria University, Newcastle upon Tyne, United Kingdom, **6** Newcastle upon Tyne Hospitals NHS Foundation Trust, Newcastle upon Tyne, United Kingdom, **7** Department of Orthopaedic Oncology, The London Sarcoma Service, Royal National Orthopaedic Hospital NHS Trust, Stanmore, United Kingdom

* s.furtado@ucl.ac.uk

**Data Availability Statement:** All relevant data are within the Submitted Manuscript and its Supporting Information Files.

## Abstract

### Background

Physical limitations are frequent and debilitating after sarcoma treatment. Markerless motion capture (MMC) could measure these limitations. Historically expensive cumbersome systems have posed barriers to clinical translation.

### Research question

Can inexpensive MMC [using Microsoft Kinect™] assess functional outcomes after sarcoma surgery, discriminate between tumour sub-groups and agree with existing assessments?

### Methods

Walking, unilateral stance and kneeling were measured in a cross-sectional study of patients with lower extremity sarcomas using MMC and standard video. Summary measures of temporal, balance, gait and movement velocity were derived. Feasibility and early indicators of validity of MMC were explored by comparing MMC measures i) between tumour sub-groups; ii) against video and iii) with established sarcoma tools [Toronto Extremity Salvage Score (TESS)), Musculoskeletal Tumour Rating System (MSTS), Quality of life-cancer survivors (QoL-CS)]. Statistical analysis was conducted using SPSS v19. Tumour sub-groups were compared using Mann-Whitney U tests, MMC was compared to existing sarcoma measures using correlations and with video using Intraclass correlation coefficient agreement.

**Funding:** The institution of one or more of the authors (CG, SF) has received, during the study period, funding from the Children with Cancer UK Charity (Grant number: 2012/139) https://www.childrenwithcancer.org.uk/childhood-cancer-info/we-fundresearch/projects-we-fund/assessing-physical-function/, Sarcoma UK Charity (Grant number: 007.2012), and Shear's Foundation and Research and Capability (RCF). The funders had no role in study design, data collection and analysis, decision to publish, or preparation of the manuscript.

**Competing interests:** The institution of one or more of the authors (AG, LR) has received, during the study period, funding from the National Institute for Health Research (NIHR) Newcastle Biomedical Research Centre and Unit based at Newcastle upon Tyne Hospitals NHS Foundation Trust and Newcastle University. The institution of one or more of the authors (AG, LR) has received, during the study period, funding from the NIHR Newcastle Clinical Research Faculty infrastructure funding. The institution of one or more of the authors (CG, SF) has received, during the study period, funding from the Children with Cancer UK Charity, Sarcoma UK Charity, and Shear's Foundation and Research and Capability (RCF). Each author certifies that his or her institution approved the human protocol for this investigation and that all investigations were conducted in conformity with ethical principles of research. This work was performed at the major clinic sites and Human Movement Room at the North of England Bone and Soft Tissue Tumor Service, Newcastle Upon Tyne Hospitals NHS Foundation Trust, UK.

## Results

Thirty-four adults of mean age 43 (minimum value—maximum value 19–89) years with musculoskeletal tumours in the femur (19), pelvis/hip (3), tibia (9), or ankle/foot (3) participated; 27 had limb sparing surgery and 7 amputation. MMC was well-tolerated and feasible to deliver. MMC discriminated between surgery groups for balance ($p<0.05*$), agreed with video for kneeling times [ICC = 0.742; $p = 0.001*$] and showed moderate relationships between MSTS and gait ($p = 0.022*$, $r = -0.416$); TESS and temporal outcomes ($p = 0.016*$ and $r = -0.0557*$), movement velocity ($p = 0.021*$, $r = -0.541$); QoL-CS and balance ($p = 0.027*$, $r = 0.441$) [* = statistical significance]. As MMC uncovered important relationships between outcomes, it gave an insight into how functional impairments, balance, gait, disabilities and quality of life (QoL) are associated with each other. This gives an insight into mechanisms of poor outcomes, producing clinically useful data i.e. data which can inform clinical practice and guide the delivery of targeted rehabilitation. For example, patients presenting with poor balance in various activities can be prescribed with balance rehabilitation and those with difficulty in movements or activity transitions can be managed with exercises and training to improve the quality and efficiency of the movement.

## Significance

In this first study world-wide, investigating the use of MMC after sarcoma surgery, MMC was found to be acceptable and feasible to assess functional outcomes in this cancer population. MMC demonstrated early indicators of validity and also provided new knowledge that functional impairments are related to balance during unilateral stance and kneeling, gait and movement velocity during kneeling and these outcomes in turn are related to disabilities and QoL. This highlighted important relationships between different functional outcomes and QoL, providing valuable information for delivering personalised rehabilitation. After completing future validation work in a larger study, this approach can offer promise in clinical settings. Low-cost MMC shows promise in assessing patient's impairments in the hospitals or their homes and guiding clinical management and targeted rehabilitation based on novel MMC outcomes affected, therefore providing an opportunity for delivering personalised exercise programmes and physiotherapy care delivery for this rare cancer.

## 1. Introduction

Multi-modal management for lower extremity musculoskeletal tumours (bone and soft tissue) includes chemotherapy, radiotherapy and surgery [1, 2], side-effects of which are wide-ranging physical deficits [3–6]. Functional outcome assessments are crucial to assess the impact of treatment as well as the response to rehabilitation interventions [7]. Established sarcoma assessments including; a disability-specific questionnaire 'Toronto Extremity Salvage Scale (TESS)' comprising of 30 questions about the difficult in performing activities of daily living [4] and a clinician-reported physical impairment tool 'Musculoskeletal tumour rating system (MSTS) [8, 9] comprising of 7 items such as range of motion (ROM), muscle strength, pain, functional activity, joint instability, deformity and emotional acceptance of the limb have inherent limitations [7]. For instance: these do not capture important objective information about postural control (balance), gait and movements related to activities of daily living [5, 10].

Clinical tools for assessing postural control and movement range from simple, time-based assessments through to full-body kinematic and kinetic examinations [11]. Simple assessments provide useful information about body stability in space, but are prone to ceiling effects and may not accurately quantify postural control strategies [11, 12].

Adding more advanced data collection tools such as force platforms and three-dimensional (3D) camera systems allows for a more detailed analysis. In addition to measuring limb movement during a functional reach test, a 3D camera system can capture spatiotemporal measures of movement trajectories and kinematics [13]. This could be particularly useful in determining patient-specific movement and stability techniques which could uncover maladaptive strategies for maintaining balance and movement [14]. A major disadvantage of these systems however is that they require multiple cameras, a large space and skin-based markers. They are also cumbersome to house and transport, expensive and need technical expertise to use and interpret. These factors limit their use to major clinical centers and research laboratories.

Markerless motion capture (MMC) can overcome limitations of traditional systems as they have the potential to remove the possible influences of body-mounted markers on people and can also promote assessments both in the clinic and natural environments [15–18]. MMC using depth-sensors have gained popularity due to advantages such as, (1) are inexpensive compared to expensive laboratory systems like the Vicon 3D motion capture, (2) are portable and (3) can perform accurate 3D tracking [19–21]. One such sensor been used more over recent years to perform MMC is the Microsoft Kinect™ (Kinect Version 1) device, developed for gaming with the Microsoft Xbox One console [19]. Kinect uses video and infra-red cameras to create a 3D map of the capture volume [19]. An inbuilt randomised decision forest algorithm automatically determines anatomical landmarks on the body, including joint centers, in close to real time [22]. In the past decade, there has been growing interest in using the Kinect for general purpose motion capturing (MoCap) [23, 24]. It has shown fair to good accuracy in assessing; postural control [21], functional movements [15], 3D position in a workplace environment [16] and classifying dance gestures when compared to gold standard assessments. Furthermore, other areas where depth sensors have been used are for indoor depth mapping applications [19]. Kinect has been used for functional movement analysis including the assessments of spatial-temporal and kinematic variables of gait and staircase ascent and descent against gold standard assessments of traditional marker-based Motion Analysis systems like Vicon 3D motion capture [23–27]. As Kinect has shown promising results in the literature, in this study, the depth-sensor 'Microsoft Kinect™' (Kinect v1) was therefore selected as the technology of choice to perform MMC.

The overall objective of this study therefore was to establish the feasibility and explore early indicators of validity of inexpensive 3D MMC to measure clinically relevant movement in patients treated for lower extremity musculoskeletal cancer. Specific research questions were: Does MMC (1) demonstrate feasibility of use and acceptability [28] in a clinical setting? (2) produce clinically useful data comparable to the literature (an indicator of face validity) [29] (3) distinguish between tumour sub-groups (an indicator of discriminant validity) [30] and (4) relate to existing clinical measures (an indicator of convergent validity) [30] and standard manual techniques (an indicator of concurrent validity) [31]?

## 2. Materials and methods

### 2.1 Patient group

The study was approved by the National Research Ethics committee (Reference: 13/NE/0296) and the Newcastle Upon Tyne Hospitals NHS Foundation Trust, R&D department (Reference: 6801).As this study involves participation by human research participants, the data sharing is

therefore bound by the National Research Ethics Committee (NREC) approval for the study which we need to abide by.

Eligible patients were approached and recruited using convenience sampling from clinics and databases between 01.02.2014 and 31.10.2014. The data was accessed for research purposes between 31.10.2014 and 21.07.2021 for processing and analysing the raw data, conducting statistical analysis and dissemination purposes e.g. writing-up reports, presenting at conferences and submitting to journals. After successfully publishing two key papers from this large project, on the topics of the use of triaxial accelerometer-based body worn monitor technology in the clinic [10] and in the community [32] respectively; this current paper which investigated a different dataset of MMC outcomes to answer questions about the feasibility and clinical applicability of the use of depth sensors for MMC was compiled. The current paper is different from the previous two publications, as it investigates the MMC approach which has never been done before for this cancer group.

Patients were enrolled into the study using Informed Consent. All patients provided written informed consent. Patients included were more than one-year post-surgery for a lower extremity musculoskeletal tumour and free of active disease or active treatment. As this was a pilot and feasibility study, the sample size established for the study was 40 patients as pilot studies recommend including over 30 patients [33]. The study included both children and adults but for the purpose of investigating MMC, 34 adult datasets were used for final analysis [33]. Our pilot study is a feasibility study exploring the feasibility of a new approach MMC for this cancer population and is of a descriptive nature, so can give an indication about what is going on with the applicability of this new approach for this patient group. We are not concerned about the representativeness of this pilot group using convenience sampling, as it is only giving us an idea about the feasibility of a new approach. The data collected and tested using statistical analysis will give us trends/early results or indicators for the next larger study which will be appropriately powered and this larger study will use probability sampling to test hypothesis rigorously. Additional details about the approach to sampling and recruitment can be found in a previous publication [10].

## 2.2 Study design

Cross Sectional Pilot and Feasibility Study.

## 2.3 Equipment

The MMC system, Microsoft Kinect™ is a motion sensor which captures functional movements in 3D space. The Kinect sensor version 1 includes skeletal tracking software which estimates the position of anatomical landmarks, such as joint centers, in close to real time [34] at a 30 Hz and 640 x 480 px spatial and depth resolution. Data were obtained from the Kinect using the Microsoft software development kit (SDK). Skeleton joint position data were obtained in three axes (converted to X = mediolateral, Y = anteroposterior, Z = vertical) from patients (Supplementary material: S1A and S1B Fig).

## 2.4 Tests performed

3D MMC Methodology: Participants stood facing the Kinect sensor at a distance of 3 metres (m), adequate to collect accurate data without causing interference [16]. The sensor was placed 1 m from the floor, with the lens perpendicular to the ground, pointing towards patients. The tests included: (1) the walk component of a 7-metre timed up and go (TUG) test, (2) unilateral stance (UL), (3a) stand to kneel (STK) and (3b) kneel to stand (KTS). A researcher stood beside the Kinect to demonstrate movements and in some instances closer to the patient to

ensure their safety. Patients who underwent an amputation were offered bilateral support if they wished to perform the test on the prosthetic leg. For those who did not wish to perform the test on their prosthetic leg, their wishes were respected. Feedback forms were used to capture patient experiences about MMC using Kinect on acceptability, user-friendliness and comfort. Each patient completed a feedback form which asked whether patients found MMC acceptable (Yes/No), user-friendly (Yes/No) and comfortable (Yes/No). Patients were also given the opportunity to write open comments about their experience of the depth sensor.

## 2.5 Assessment using video

Temporal measures of STK and KTS test was calculated using observational video analysis and MMC. The start and finish of each activity was selected based on the start and the end of the movement of the head. Video derived time was not selected for other tests (Unilateral stance and TUG test) because the temporal measures captured during these tests were not indicative of the time taken to complete the activity. On the other hand, temporal measures of KTS and STK indicated the time taken to complete kneeling and rising from kneeling activity. MMC versus Video was therefore compared for only STK and KTS activities, to explore early indicators of concurrent validity.

## 2.6 Tumour sub-groups

Patients were grouped by tumour type [bone tumour (BT) or soft tissue sarcoma (STS)] and surgery [limb sparing (LSS) or amputation (AMP)]. MMC outcomes were compared between tumour groups to explore early indicators of discriminant construct validity [30].

## 2.7 Clinic scales in sarcoma

Traditional sarcoma measures; disability (TESS) [4], impairment (MSTS) [8, 9], quality of life (Quality of life-Cancer survivors (QoL-CS) [35] (Table 1) were collected. Clinical scores were collected to compare validated clinical tools in the sarcoma population (e.g TESS) against MMC; to investigate whether clinically sensible relationships were observed between the measures. This was assessed using the ICF biopsychosocial health model [36, 37] to explore whether sensible clinically sensible relationships existed, to explore early indicators of convergent construct validity [30].

## 2.8 Processing of data obtained from MMC

Raw MMC data was screened and activity-stamped visually using Microsoft Excel 2013 (v15.0). Following this, data processing used anatomical landmark displacements during tests. Raw data from walk, unilateral stance, STK and KTS (Supplementary material: S2A–S2C Fig) were processed to obtain MMC measures by activity (Table 1, Supplementary material: S2A–S2C Fig). Outcome measures obtained from MMC were then classified by known functional domains as follows [15, 21, 24–26] and used for final analysis.

(a) **Temporal:** Time-based measures recorded during unilateral stance, STK and KTS performance measured in seconds

(b) **Balance:** Amplitude of shift from midline and standard deviation were used as balance outcomes keeping with standard clinical outcomes. Balance outcomes were collected during the unilateral stance, STK and KTS tests measured in metres (m). During unilateral stance, the knee marker was used to capture the start and the end of the test and the movement of the head marker was utilised to measure postural sway. For the STK and KTS tests, the movement of the head marker was used to determine the extent of postural sway from midline.

**Table 1. Existing clinical measures and MMC measures [obtained using Microsoft Kinect[TM]] for patients with tumours.**

| S. No | Clinic measures | Sub-domains/What does the outcome measure capture | Outcomes | Scores |
|---|---|---|---|---|
| *Existing clinic measure* | | | | |
| 1. | Toronto Extremity Salvage Score (TESS) (4) | 30-item reported by patients | Physical disability | Scores range from 0 to 100 (worst to best outcomes) |
| 2. | Quality of Life for Cancer Survivors (QoL-CS) (23) | 41-item questionnaire | QoL | Scores range from 0 to 100 (worst to best outcomes) |
| 3. | Musculoskeletal Tumour Society score (MSTS) version developed in 1987 (MSTS-1987) for the Lower Limb (8) | 7 sub-domains range of motion, stability, deformity, pain, muscle strength, functional activity and emotional acceptance | Impairment | The MSTS total score is expressed from 0–35 (worst to best physical functioning). Individual sub-domain score is 0–5 |
| *Outcome measures derived from MMC* | | | | |
| *Test 1*: Walk component of Timed Up and Go Test (TUG) (Repeat x3): Patients were asked to complete a timed up and go test with Kinect recording the walking component of test. Horizontal displacement of the head marker was used to derive outcome measures | | | | |
| 1. | Walk Distance (m) | Walk distance recorded by MMC | Spatial gait | A lower walk distance compared to healthy individuals reflects impaired gait |
| 1. | Walk Time (s) | Walk time to cover the distance | Temporal gait | A higher walk time compared to healthy individuals reflects impaired gait |
| 2. | Walk Velocity (m/s) | Walk velocity during this distance | Spatio-temoral gait | A lower walk velocity compared to healthy individuals reflects impaired gait |
| *Test 2*: Unilateral stance for up to 30 seconds on affected and unaffected limb (Repeat x3): Patients started from a position of standing with eyes open. Patient was asked to lift one leg off the floor without support if able. If patient was unable to perform the test without support, the test was performed with support and this was recorded. Vertical displacement of the knee marker was used to identify the start and end of the test and the movement of the head marker was utilised to derive outcome measures | | | | |
| 1. | Unilateral stance total time (s) | Time for which patients' could perform the unilateral stance test | Temporal measure | A lesser time compared to healthy individuals reflects impaired balance [38] |
| 2. | Unilateral stance Anterior-posterior range (m) | Distance a patient swayed in the AP direction during their unilateral stance test | A-P Balance | A higher range of sway compared to healthy individuals reflects impaired balance [21] |
| 3. | Unilateral stance Medio-lateral range (m) | Distance a patient swayed in the ML direction during the unilateral stance test | M-L Balance | A higher range of sway compared to healthy individuals reflects impaired balance [21] |
| 3. | Unilateral stance Antero-posterior sd (m) | Standard deviation (sd) of change of AP range | A-P Balance change | A higher range of sd compared to healthy individuals reflects impaired balance [21] |
| 4. | Unilateral stance Medio-lateral sd (m) | Standard deviation (sd) of change of ML range | M-L Balance change | A higher range of sd compared to healthy individuals reflects impaired balance [21] |
| *Test 3a*: Stand to kneel test (STK, Kneeling Activity) (Repeat x3): Patients started the test from a position of standing with eyes open. Patients were asked to kneel on a firm kneeling mat without support where able. Vertical displacement of the head marker was used to derive outcome measures | | | | |
| 1. | STK Total time (s) | Time taken to perform stand to kneel test | Temporal measure | The ability to complete kneeling is a good outcome [39]. However a higher total time compared to healthy individuals reflects an impaired activity [39] |
| 2. | STK Peak velocity (m/s) | Maximum velocity attained during stand to kneel test | Spatio-temporal | A lower peak velocity compared to healthy individuals reflects poor balance/postural control during kneeling [40] |
| 3. | STK Anterior amplitude (m) | Maximum anterior amplitude attained during kneeling activity | A-P Balance | Similar to unilateral stance reasoning [21], a higher amplitude of sway compared to healthy individuals reflects an impaired balance/postural control during kneeling. |
| 4. | STK Lateral amplitude (m) | Maximum lateral amplitude attained during kneeling activity | M-L Balance | A higher amplitude of sway compared to healthy individuals reflects an impaired balance/postural control during kneeling, as the participant is moving out of their base of support. |
| *Test 3b*: Kneel to stand test (KTS, Rising from Kneeling Activity) (Repeat x3): Patients started the test from a position of kneeling. Patients were asked to stand from kneeling, without support where able. Support was provided when required and this was recorded for Tests 3. Vertical displacement of the head marker was used to derive outcome measures | | | | |
| 1. | KTS Total time (s) | Time taken to perform kneel to stand test | Temporal measure | A higher total time compared to healthy individuals reflects an impaired activity |

*(Continued)*

**Table 1.** (Continued)

| S. No | Clinic measures | Sub-domains/What does the outcome measure capture | Outcomes | Scores |
|---|---|---|---|---|
| 2. | KTS Peak velocity (m/s) | Maximum velocity attained during rising from kneeling activity | Spatio-temporal measure | A lower peak velocity compared to healthy individuals reflects poor balance/postural control during rising from kneeling [40] |
| 3. | KTS Anterior amplitude (m) | Maximum anterior amplitude attained during kneel to stand test | A-P Balance | A higher amplitude of sway compared to healthy individuals reflects an impaired balance/postural control during kneeling |
| 4. | KTS Lateral amplitude (m) | Maximum anterior amplitude attained during during kneel to stand test | M-L Balance | A higher amplitude of sway compared to healthy individuals reflects an impaired balance/postural control during kneeling. |

(c) **Gait:** Walking distance, time and velocity obtained from walk component of 7-metre TUG test and was measured in m, s and m/s.

(d) **Movement velocity:** Movement velocities recorded during STK and KTS measured in m/s. Movement velocity is defined as vertical velocity. The start point and the end point were defined by the movement of the head from kneeling to stand and stand to kneel.

## 2.9 Clinical interpretation of good versus poor clinical outcomes

The clinical interpretation of poor and good outcomes have been detailed in Table 1.

## 2.10 Skeletal model tracking

The MMC system, Kinect SDKs provide a skeleton tracker of up to 20 body joints and code samples that can be used to track movement. Patient movements can be visualised without using video. The skeleton tracker classifies each pixel of depth images as components of a joint using trained decision forests [22].

## 2.11 Statistical analysis

Statistical analysis was carried out using SPSS v19 (IBM). Parametric data were expressed using means and standard deviations (SDs) (minimum value—maximum value) and non-parametric data using medians with interquartile ranges (IQR). Outcomes obtained using MMC were compared between tumour sub-groups using Independent t or Mann-Whitney U tests. We used the Bonferroni correction to address correction for multiple measures for the between group comparisons and set the alpha level at 0.05/6 = 0.008. Pearson and Spearman's rho correlations were used to investigate relationships between measures obtained from MMC and existing measures. Intraclass correlation coefficient (ICC) agreement, two-way random effects model and Bland Altman analysis tested agreement between MMC measures and standard manual techniques. ICC, is a descriptive statistic which was used to describe the strength of units in the same group and their resemblance with each other. We interpreted ICC agreements as: poor ($< 0.5$), moderate (between 0.5 and 0.75), good (0.75 to 0.9) and excellent ($> 0.9$) (30, 31). Significance was taken at the 0.05 level.

## 3. Results

### 3.1 Demographic and clinical characteristics

34 adults of mean age 43 ± 20 years participated. Patients were treated for BT (n = 21) or STS (n = 13) in the femur (n = 19), pelvis/hip (n = 3), tibia (n = 9), or ankle/foot (n = 3). 27 underwent LSS and 7 AMP. Median time from surgery was 79 months (33–108). 15/34 patients

received chemotherapy, and 13/34 received radiotherapy. Further details about potentially eligible patients approached, those who refused and participated can be found in the recent publication [10]. A total of 97 patients were approached in this study, 65 patients satisfied all eligibility criteria and 32 patients were ineligible. Out of the 65 patients, 21 patients declined participation and 4 patients were non-contactable, leaving us with 40 patients in the study out of which 34 were adults and their data were used for final analysis. Patients who declined participation were mainly those who lived far away from the specialist centre where assessments were undertaken and did not prefer to travel these distances to participate in the research.

## 3.2 Feasibility, data loss and acceptability of MMC in the clinic

MMC was feasible to perform using Kinect and was quick to set up, taking approximately 10 minutes. Data downloading and processing were straightforward taking approximately 20 minutes to complete. Of 34 participants, one had a hind quarter amputation and could not perform any clinic tests. Furthermore, the MMC system did not record data for 3 patients due to a technical issue with data collection. Of the remainder, 8 were either unable to or refused to perform kneeling and 2 unilateral stance on the affected side. During data processing, 2 trials data (one kneeling and one unilateral stance) were invalid and removed from final analysis. Therefore of 33 patients assessed, data related to 30 walking, 21 kneeling, 27 unilateral stance on affected side and 30 unilateral stance on unaffected side test datasets were available for analysis.

Of patients who returned feedback forms with questions answered about the MMC system (n = 19), patients who found the MMC approach in the clinic: acceptable for use: 19/19 (100%) and comfortable: 19/19 (100%) after sarcoma treatment.

## 3.3 Early Indicators of Validity of MMC in musculoskeletal cancer patients

Outcome measures derived from MMC are summarised in Table 2.

**3.3.1. Outcomes from MMC in tumour sub-groups.** The MMC approach distinguished between BT and STS groups and the LSS and AMP groups for temporal and balance outcomes respectively in the unaffected limb (p<0.05*) (Table 2). For instance, the Unilateral Stance total time (s) of unaffected limb was found to be higher in patients operated for bone tumours 29.69 (25.56–30.00) s than those operated for soft tissue tumours 26.61 (15.88–29.49) s, indicating that patients with bone tumours could stand on their unaffected limb for longer than those with soft tissue tumours. Furthermore the Unilateral stance A-P range (m) and M-L range (m) of unaffected side and the Unilateral stance A-P sd (m) and M-L of unaffected side were found to be significantly higher in patients who had an amputation compared to those with limb sparing surgeries (p<0.05*) (Table 2).

**3.3.2. Outcome measures from MMC versus clinical scales.** Median (range) TESS score was 83.6 (IQR 62.1 to 93.8 [8.3 to 100.0]), mean MSTS score 24.5 (SD 7.9 [5.0 to 35.0]), median 3-meter TUG time 10.8 seconds (IQR 8.5 to 12.7 [7.9 to 32.3]) and median QoL-CS total score 7.1 (IQR 6.1 to 7.8 [2.7 to 9.1]). Significant correlations were observed between MSTS, TESS, QoL-CS and MMC (p<0.05*) (Table 3 and Fig 1A–1H).

For instance, MSTS demonstrated significant negative correlations with KTS Anterior amplitude. This suggested that high impairments of function i.e. impacted joint ROM, muscle strength, joint stability, limb length discrepancy, pain and gait problems (indicating high functional impairments) were associated with a large postural sway during kneeling (indicating affected balance). MSTS also correlated positively with STK peak velocity and walk velocity; suggesting high MSTS scores (low functional impairments) are associated with high

**Table 2. Functional outcomes in tumour patients captured using MMC, BT vs STS, LSS vs AMP.**

| Outcome measures derived from MMC | Tumour patients (n = 33) | BT group (n = 21) | STS group (n = 12) | p-value for BT vs STS groups | LSS group (n = 27) | AMP group (n = 6) | p-value for LSS vs AMP groups |
|---|---|---|---|---|---|---|---|
| | Median/Mean Values (25th– 75th percentile, 1QR/Min-max) | Median/Mean Values (25th– 75th percentile, 1QR/Min-max) | Median/Mean Values (25th– 75th percentile, 1QR/Min-max) | | Median/Mean Values (25th– 75th percentile, 1QR/Min-max) | Median/Mean Values (25th– 75th percentile, 1QR/Min-max) | |
| **Temporal measures (assessing time taken to complete/sustain activity)** | | | | | | | |
| Unilateral stance total time (s) of affected limb | 20.38 (6.81–29.32) | 20.79 (8.27–29.46) | 11.77 (3.82–18.68) | 0.111 | 20.53 (8.27–27.62) | 3.96 (1.81–29.56) | 0.382 |
| Unilateral stance total time (s) of unaffected limb | 29.22 (22.71–29.92) | 29.69 (25.56–30.00) | 26.61 (15.88–29.49) | 0.047* | 29.36 (21.40–29.92) | 29.04 (19.30–29.86) | 0.795 |
| STK total time (s) | 2.00 (1.48–2.82) | 2.05 (1.51–2.81) | 1.90 (1.430–3.80) | 0.881 | 1.90 (1.53–2.83) | 2.24 (1.43–3.40) | 0.585 |
| KTS total time (s) | 2.83 (1.90–3.84) | 3.37 (2.06–3.82) | 2.53 (1.53–4.83) | 0.551 | 2.67 (1.77–3.80) | 3.37 (2.33–4.26) | 0.586 |
| **Balance measures (assessing the range during movements or high amplitude of shift from midline or base of support)** | | | | | | | |
| Unilateral stance A-P range (m) of affected side | 0.08 (0.05–0.11) | 0.07 (0.05–0.10) | 0.11 (0.06–0.21) | 0.088 | 0.07 (0.05–0.10) | 0.13 (0.06–0.20) | 0.177 |
| Unilateral stance M-L range (m) of affected side | 0.06 (0.05–0.09) | 0.06 (0.04–0.07) | 0.08 (0.07–0.10) | 0.059 | 0.07 (0.05–0.08) | 0.06 (0.04–0.20) | 0.706 |
| Unilateral stance A-P sd (m) of affected side | 0.02 (0.01–0.02) | 0.02 (0.01–0.02) | 0.02 (0.02–0.06) | 0.054 | 0.02 (0.01–0.02) | 0.02 (0.01–0.06) | 0.307 |
| Unilateral stance M-L sd (m) of affected side | 0.02 (0.01–0.02) | 0.01 (0.01–0.02) | 0.02 (0.02–0.03) | 0.064 | 0.02 (0.01–0.02) | 0.02 (0.01–0.06) | 0.314 |
| Unilateral stance A-P range (m) of unaffected side | 0.08 (0.06–0.12) | 0.08 (0.06–0.12) | 0.08 (0.07–0.14) | 0.564 | 0.08 (0.06–0.11) | 0.13 (0.11–0.21) | 0.006* |
| Unilateral stance M-L range (m) of unaffected side | 0.05 (0.04–0.07) | 0.05 (0.04–0.07) | 0.05 (0.04–0.07) | 0.875 | 0.05 (0.04–0.07) | 0.07 (0.05–0.16) | 0.021* |
| Unilateral stance A-P sd (m) of unaffected side | 0.02 (0.01–0.02) | 0.02 (0.01–0.03) | 0.02 (0.01–0.02) | 0.963 | 0.01 (0.01–0.02) | 0.03 (0.02–0.04) | 0.004* |
| Unilateral stance M-L sd (m) of unaffected side | 0.01 (0.01–0.02) | 0.01 (0.01–0.02) | 0.01 (0.01–0.01) | 0.476 | 0.01 (0.01–0.01) | 0.02 (0.01–0.03) | 0.014* |
| STK Anterior amplitude (m) | 0.22 (0.11–0.31) | 0.14 (0.10–0.28) | 0.31 (0.22–0.44) | 0.048 | 0.22 (0.08–0.31) | 0.23 (0.14–0.34) | 0.533 |
| STK Lateral amplitude (m) | 0.17 (0.05–0.32) | 0.18 (0.07–0.39) | 0.12 (0.03–0.26) | 0.295 | 0.12 (0.05–0.26) | 0.27 (0.04–0.41) | 0.507 |
| KTS Anterior amplitude (m) | 0.13 (0.10–0.24) | 0.13 (0.10–0.25) | 0.12 (0.08–0.25) | 0.764 | 0.12 (0.08–0.23) | 0.16 (0.13–0.34) | 0.079 |
| KTS Lateral amplitude (m) | 0.10 (0.02–0.18) | 0.11 (0.01–0.28) | 0.10 (0.01–0.17) | 0.572 | 0.12 (0.01–0.17) | 0.02 (0.02–0.37) | 0.813 |
| **(C) Gait measures (assessing parameters of gait)** | | | | | | | |
| Walk distance (m) | 2.50 (2.21–2.77) | 2.52 (2.19–2.8) | 2.43 (2.21–2.65) | 0.481 | 2.48 (2.18–2.73) | 2.60 (2.03–2.99) | 0.392 |

(*Continued*)

**Table 2.** (Continued)

| Outcome measures derived from MMC | Tumour patients (n = 33) | BT group (n = 21) | STS group (n = 12) | p-value for BT vs STS groups | LSS group (n = 27) | AMP group (n = 6) | p-value for LSS vs AMP groups |
|---|---|---|---|---|---|---|---|
| | Median/Mean Values (25th– 75th percentile, 1QR/Min-max) | Median/Mean Values (25th– 75th percentile, 1QR/Min-max) | Median/Mean Values (25th– 75th percentile, 1QR/Min-max) | | Median/Mean Values (25th– 75th percentile, 1QR/Min-max) | Median/Mean Values (25th– 75th percentile, 1QR/Min-max) | |
| Walk time (s) | 2.24 | 2.28 | 2.12 | 0.244 | 2.24 | 2.27 | 0.917 |
| | (1.84–2.57) | (1.8525–2.6475) | (1.79–2.39) | | (1.85–2.56) | (1.71–2.93) | |
| Walk velocity (m/s) | 1.10 | 1.08 | 1.26 | 0.058 | 1.09 | 1.13 | 0.959 |
| | (0.89–1.30) | (0.80–1.14) | (1.03–1.36) | | (0.91–1.31) | (0.75–1.23) | |
| **(D) Movement velocity measures (assessing the velocity during functional movements)** | | | | | | | |
| STK Peak velocity (m/s) | -0.57 | -0.59 | -0.54 | 0.278 | -0.57 | -0.59 | 0.696 |
| | [(-0.62-(0.44)] | [(-0.72-(0.42)] | [(-0.57–0.44)] | | [(-0.63 - (-0.44)] | [(-0.70 –(-0.42)] | |
| KTS peak velocity (m/s) | 0.82 | 0.84 | -0.54 | 0.247 | 0.68 | 0.93 | 0.959 |
| | (0.61–0.99) | (0.63–1.05) | [(-0.57- (-0.44)] | | (0.55–0.88) | (0.71–1.14) | |

Statistical test: Mann-Whitney U test. p-value–correlation between variables (* = statistically significant with and without Bonferroni correction

**Table 3. Relationships between established clinical scales and outcome measures derived from MMC.**

| Clinical scales in sarcoma | (A) Temporal measures | r value | p-value | (B) Balance measures | r value | p-value | (C) Gait Measures | r value | p-value | (D) Movement velocity measures | r value | p-value |
|---|---|---|---|---|---|---|---|---|---|---|---|---|
| MSTS total (impairment) | Unilateral stance of affected side total time (s) | 0.268 | 0.176 | KTS Anterior amplitude (m) | -0.567 | 0.007* | Walk distance (m) | 0.005 | 0.978 | STK peak velocity (m/s) | 0.502 | 0.020* |
| | STK Total time (s) | -0.206 | 0.370 | Unilateral stance A-P range of affected side | 0.064 | 0.751 | Walk time (s) | -0.416 | 0.022* | KTS peak velocity (m/s) | -0.030 | 0.896 |
| | KTS Total time (s) | -0.347 | 0.124 | Unilateral stance M-L range of affected side (m) | 0.140 | 0.487 | Walk velocity (m/s) | 0.511 | 0.004* | | | |
| TESS (disability) | Unilateral stance total time (s) | 0.340 | 0.113 | Unilateral stance M-L range of unaffected side (m) | 0.449 | 0.024* | Walk distance (m) | -0.045 | 0.831 | STK peak velocity (m/s) | 0.541 | 0.021* |
| | STK Total time (s) | -0.372 | 0.128 | KTS Anterior amplitude (m) | -0.564 | 0.015* | Walk time (s) | -0.237 | 0.253 | KTS peak velocity (m/s) | -0.011 | 0.964 |
| | KTS Total time (s) | -.0557* | 0.016* | STK Anterior amplitude (m) | 0.236 | 0.345 | Walk velocity (m/s) | 0.291 | 0.159 | | | |
| QoL-CS total score | Unilateral stance total time (s) | 0.130 | 0.554 | Unilateral stance M-L range of unaffected side (m) | 0.441 | 0.027* | Walk distance (m) | 0.138 | 0.511 | STK peak velocity (m/s) | 0.243 | 0.331 |
| | STK Total time (s) | -0.449 | 0.062 | STK Anterior amplitude (m) | 0.098 | 0.699 | Walk time (s) | -0.162 | 0.439 | KTS peak velocity (m/s) | -0.124 | 0.625 |
| | KTS Total time (s) | -0.341 | 0.166 | STK Lateral amplitude (m) | -0.433 | 0.072 | Walk velocity (m/s) | 0.358 | 0.079 | | | |
| QoL-CS social sub-score (QoL) | STK Total Time (S) | -0.516 | 0.028* | KTS Anterior amplitude (m) | -0.492 | 0.038* | | | | | | |
| | KTS Total Time (S) | -0.560 | 0.016* | KTS Lateral amplitude (m) | -0.522 | 0.026* | | | | | | |

Statistical test = Spearman's correlation.

*A p value < 0.05 was considered statistically significant; BWM = body-worn monitor, TUG = timed up and go test; MSTS = Musculoskeletal Tumour Society Scoring system; TESS = Toronto Extremity Salvage Score; QoL-CS = Quality of life-Cancer survivors; RMS = Root Mean Square

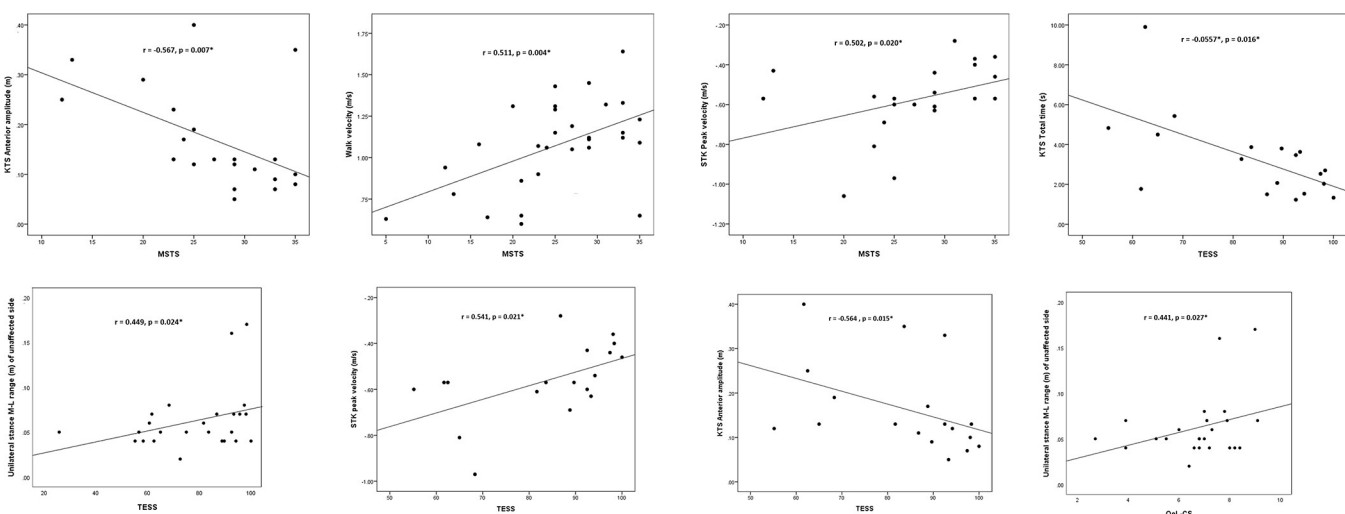

**Fig 1. Correlations of traditional sarcoma measures with outcomes captured by MMC.** Outcomes captured by MMC significantly relate to MSTS, TESS and QoL-CS (Fig 1A–1H). A: KTS Anterior amplitude significantly relates to MSTS. B: Walk velocity significantly relates to MSTS. C: STK Peak velocity significantly relates to MSTS. D:KTS Total time relates to TESS. E:Unilateral stance M-L range of unaffected side relates to TESS. F:STK Peak velocity relates to TESS. G:KTS Anterior amplitude relates to TESS. H: Unilateral stance M-L Range of unaffected side relates to QoL-CS.

movement velocity (good speed/function) during kneeling and higher walk velocity (good gait speed/function) respectively.

TESS showed significant negative correlations with KTS total time and KTS anterior amplitude, suggesting high levels of disability (low TESS) were associated with more time to execute KTS (poor function) and a high KTS anterior amplitude i.e. larger postural sway (indicating affected balance) during kneel to stand activity. TESS also showed significant positive correlations with Unilateral stance M-L range of unaffected side and STK peak velocity, suggesting that high levels of disability (low TESS) are associated with a low postural sway of the unaffected side during unilateral stance (good function on the unaffected side) and low movement velocity during STK (poor function), reflecting important associations.

QoL-CS showed significant positive correlations with Unilateral stance M-L range of unaffected side, suggesting that a reduced postural sway (impaired balance) on the unaffected side is associated with a reduced QoL. Whereas the QoL social sub-scales showed significant negative correlations with KTS Anterior amplitude and KTS Lateral amplitude. This suggested that larger posture sways in the anterior and lateral directions during KTS were associated with a poor QoL from a social integration perspective.

**3.3.3. Agreement of MMC with video for temporal accuracy.** STK time captured by video [2.00 +/- 0.67 seconds] showed moderate agreement with STK time captured by Kinect [1.98+/- 0.90 seconds] (ICC agreement = 0.742; p = 0.001*) (Table 4). Bland-Altman analysis (Fig 2A) demonstrated that Kinect underestimated values with a bias of 0.014 in comparison to the video. The 95% limits of agreement were +1.44 seconds and -1.41 seconds.

Similarly, KTS time captured by Kinect [1.73 +/- 0.60] (ICC agreement 0.624; p = 0.010*) showed moderate agreement with KTS time by video [1.62 +/- 0.47] (Table 5). Bland-Altman analysis (Fig 2B) demonstrated that Kinect over-estimated the values with a bias of -0.112 in comparison to video, with 95% limits of agreement of +0.98 seconds and -1.21 seconds.

**3.3.4. Case studies of skeletal model tracking.** Skeletal tracking data captured for Case study A during the unilateral stance test and Case study B during the STK and KTS test (Supplementary material: S1 Table) demonstrated that it is feasible to collect and visually inspect movements or activities from anonymised data collected; in patients with sarcomas.

**Table 4. ICC agreement for STK time captured by video and MMC.**

| Test (n = 25) | Mean | SD | ICC average measures | 95% confidence interval | | p value |
|---|---|---|---|---|---|---|
| | | | | Lower bound | Upper bound | |
| Kinect_STK_time(s) | 1.98 | 0.90 | 0.74 | 0.41 | 0.89 | 0.001* |
| Video_STK_time(s) | 2.00 | 0.67 | | | | |

Cronbach's alpha = 0.734

Cronbach's alpha based on standardized items = 0.754

p value–agreement between devices (*statistically significant); two-way random effects model where both people effects and measures effects are random; †the estimator is the same, whether the interaction effect is present or not; ‡type A intraclass correlation coefficients using an absolute agreement definition; ICC = intraclass correlation coefficients.

# 4. Discussion

## 4.1. Feasibility and acceptability of MMC

This is the first pilot to investigate MMC in patients with musculoskeletal cancers using a low-cost portable depth-sensor. Overall, the MMC approach was found to be feasible to deliver, straightforward to collect and acceptable to use and comfortable by those who completed the feedback survey. MMC also confirmed early indicators of validity for certain measures and sensible clinical trends were present in others which was promising. The positives were that MMC captured temporal measures, balance, gait and movement velocities outcomes promptly, with minimal data loss. MMC data could also assess slowness of activities in patients, postural control strategies, gait and movement velocities alterations, not obtainable routinely but which add novel knowledge to guide rehabilitation and treatment after major surgery for this rare cancer.

## 4.2. Early Indicators of validity of MMC

**4.2.1. Comparison to healthy individuals from reference literature.** Outcomes derived from MMC made broad clinical sense and some variables were comparable to published literature [21, 24]. Unilateral stance anterior-posterior (A-P) range (m) of the affected side [0.08 (0.05–0.11) m] and, unilateral stance medio-lateral (M-L) range (m) of affected side [0.06 (0.05–0.09) m], were higher than balance values reported by Clark et al for healthy individuals [0.051 (0.032) m] [21]. The closest anatomical marker in Clark's study which compared to our

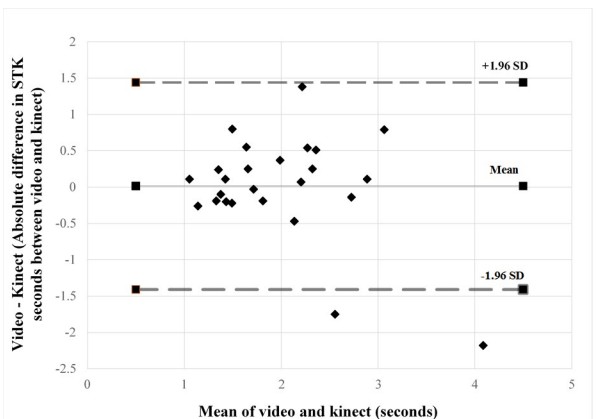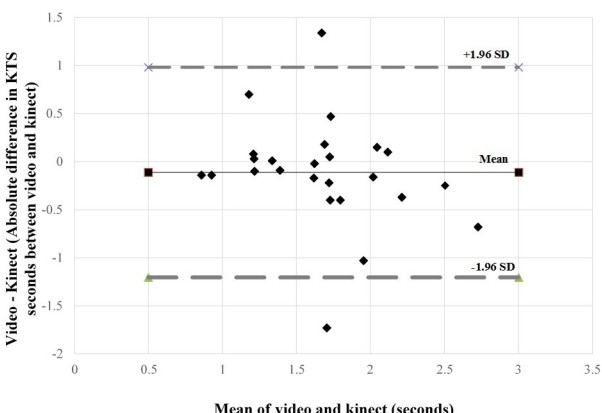

**Fig 2.** A: Agreement of Kinect and video times for STK. B: Agreement of Kinect and video times for KTS.

**Table 5. ICC agreement for KTS time captured by video and MMC.**

| Test (n = 25) | Mean | SD | ICC average measures | 95% confidence interval | | p value |
|---|---|---|---|---|---|---|
| | | | | Lower bound | Upper bound | |
| Kinect_STK_time(s) | 1.73 | 0.59 | 0.62 | 0.16 | 0.83 | 0.010* |
| Video_STK_time(s) | 1.62 | 0.47 | | | | |

Cronbach's alpha = 0.624

Cronbach's alpha based on standardized items = 0.635

p value–agreement between devices (*statistically significant); two-way random effects model where both people effects and measures effects are random; †the estimator is the same, whether the interaction effect is present or not; ‡type A intraclass correlation coefficients using an absolute agreement definition; ICC = intraclass correlation coefficients

study's head marker was the sternum marker assessing balance in unilateral stance. Although there was a difference in the protocol of both studies, patients treated with sarcoma presented with a higher postural sway in the A-P direction while standing on their affected limb [0.08 (0.05–0.11) m] compared to healthy individuals [0.06 (0.02) m] in Clark's study. This demonstrates that patients treated with sarcoma demonstrate reduced postural control than healthy individuals putting them at a higher risk of falls, which can be attributed to postural control changes in the affected side after major surgical resections. A-P sway was higher compared to M-L sway which highlight the postural compensatory mechanisms in people operated for sarcomas.

In previous studies, balance and postural control strategies during quiet standing were found to be affected in patients treated for sarcoma [5, 10] however postural control strategies during other activities were never investigated. This is the first study demonstrating that postural control strategies measured objectively are affected during unilateral stance and kneeling and require targeted rehabilitation e.g. balance exercises to improve postural control during such activities.

Similarly for gait outcomes, our study patients showed gait velocity [1.10 (0.89–1.30) m/s] was lower than the mean gait velocity [1.26±0.12 m/s] in healthy adults from another study [24]. This could be explained on the basis that our study patients had large surgeries on their lower limbs which affect locomotion and gait.

**4.2.2. Comparison between tumour sub-groups.** In our study, the MMC approach distinguished between BT and STS groups, and these comparisons were made as BT are often deep seated compared to STS which are usually more superficial and this often clinically results in differences between BT and STS groups. However as this was pilot and feasibility work, the extent of surgery and reconstruction were not collected. In future work, this will be considered as the size of the tumour and extent of surgery plays a big role alongside the primary diagnoses of the disease. For example, the functional outcome may be the same in large sized sarcomas whether primarily affecting bone or soft tissues.

MMC distinguished between LSS and AMP groups for temporal and balance outcomes in the unaffected limb (p<0.05*). Amputees demonstrated a significantly higher unilateral stance A-P range (m) and M-L range (m) of the unaffected side compared to the LSS group (p<0.05). This could be explained on the basis of alterations in the sensory and proprioceptive inputs after a major limb loss [41, 42] potentially leading to significant alterations and compensations in the unaffected side. This suggests that unaffected side testing is a sensitive test for patient groups.

**4.2.3. Outcome measures derived from MMC versus clinical scales.** Our study revealed important associations between MMC and established clinical measures. Higher impairment

was associated with poor balance, gait and movement velocity, which in turn, were significantly associated with worse disability and QoL. This emphasises the importance of using MMC, as findings revealed transition to and from floor activities, inherently known to be challenging for this patient group [3, 4, 43], is associated with a poor QoL. This novel information highlights relationships between outcomes and can guide the delivery of rehabilitation. For instance, in order to improve QoL of patients, people's impairments such as joint range of motion, muscle strength, balance and gait are important to treat using targeted exercise prescriptions. MMC alongside clinical scores provide sensible relationships and comprehensive information about the patient's functional status which is not traditionally captured.

**4.2.4. Agreement of MMC with manual techniques.** Galna et al in 2014 reported that MMC consistently over- or under-estimated some measures of spatial and temporal movements compared to a 3D marker-based motion analysis system, dependant on the specific movement [15]. This study in people was in people with Parkinson's. Our results agree with this study, as MMC in our study underestimated the STK total time and overestimated the KTS and showed moderate agreements. Calibrations might address such biases from MMC systems, along with future software updates which might help in overcoming some of the observed differences. Addressing biases could minimise errors and facilitate MMC to become a promising clinical assessment for risk-based screening programs in clinics and homes of patients.

## 4.3. Strengths, limitations, clinical implications and future work

**4.3.1. Strengths.** This study is invaluable as a first step, to represent the use of this novel approach in this patient group. This pilot work tested the applicability of MMC across a wide range of the heterogenous population and demonstrated, which represents generalisability to all sub-groups. Patients who responded to the feedback survey had positive feedback about MMC which was promising.

**4.3.2. Limitations.** The authors are aware that production of the depth-sensor used in this study has stopped but as technology evolves newer systems for performing MMC will be available. An important message from this first study of its kind; is MMC feasibility and its agreement with standard clinical methods which when supported with future research can facilitate its translation into busy clinical settings. Although. we particularly used the head and knee anatomical landmark displacements, a more comprehensive approach to detail postural control strategies by Clark et al. would be helpful. This approach looks at multiple points such as pelvis center, hip, hand, elbow, shoulder, sternum, ankle displacement data streams to derive quantitative data [21], to assess adaptive mechanisms, postural reactions and compensations for a complex patient group of sarcomas.

Another limitation was that unilateral stance with eyes closed was not performed, and so we are unable to comment on whether MMC can detect adaptive mechanisms performed by patients to compensate when vision is removed. Despite two reminders sent to patients, the response rate for the feedback points was found to be only 50% (19/34). This can introduce bias about acceptability and user-friendliness of MMC and results need to be treated with caution. In a larger study, we will approach this issue by completing feedback forms with patients in clinic or on phone, rather than by post to ensure a good response rate is achieved.

Although this was a pilot sample of convenience and have its inherent limitations; the results of this study highlighted important results about feasibility. Being a pilot study, patient groups were required to be clubbed together to have sufficient numbers in sub-groups e.g. amputation and therefore interpretation about individual clinical groups e.g. above knee amputation or below knee amputation is not possible. This will be addressed in future work.

where patients will be classified into homogenous groups for analysis and also healthy controls will be included for comparison. Use of convenient sampling is a weakness of our study, and the prevalence found in our study may differ from that in the full target population due to possible selection bias. To address this, for the larger future study more appropriate approaches to sampling e.g. purposive sampling will be used.

**4.3.3. Clinical implications.** MMC can be used as a screening tool to identify patients with impaired balance, gait and movement control. Personalised balance and gait exercise programmes and introducing challenges like eyes closed, perturbations and changes in surfaces can improve movement restoration, agility and control of movements, preventing trips and falls, minimising injury, increasing confidence and reducing dependence; ultimately improving the QoL. The Skeletal tracking feature is useful for assessing movement quality and postural control strategies [21] in a way that effectively anonymises data and provides feedback to clinicians and patients to reduce movement compensations. The goal is to use such cost-effective systems as rehabilitation tools in this cancer group and explore its transferability across other cancers and orthopaedic surgery groups.

**4.3.4. Future work.** The next steps would be to conduct a larger study with sufficient power using purposive sampling strategy to validate the MMC approach. In future, other methods such as stopwatch, wearables like accelerometery in the clinic [10] and community [32] or fitbits, IMUs, mobile apps or Xsens could be utilised to collect outcomes alongside MMC. Xsens will be a useful comparison to MMC, if joint range of motion is the area of focus. MMC, wearables and Vicon 3D motion capture could collect outcomes such as movement velocities, balance and temporal parameters, which can allow a cross validation work between portable cost-effective devices and Vicon 3D motion capture.

Another future step is that, the MMC protocols from this study could also be deployed in future larger multi-centre studies in sarcoma sub-groups for example, distal femoral replacements, proximal femoral replacements, pelvic resections, pelvic reconstructions to perform further gold standard validation work as described earlier. We aim to include an assessment of external validity within the next future multi-centre work to increase generalisability of the results. Using MMC systems e.g. the newer versions of Kinect with the Microsoft Xbox One console could be attractive if used for exergaming for this cancer population, especially popular amongst children and young people. We will also utilise MMC in future studies to remotely monitor falls and gait of patients at home [27, 44] and this could be particularly useful in global pandemic conditions e.g. during the recent covid-19, where remote and digital healthcare remains the best means to avoid transmission or spread of the virus and deliver better healthcare [45].

## 5. Conclusion

The MMC approach demonstrated feasibility and early indicators of certain types of validity in capturing novel information about temporal, balance, gait, movement velocity and skeletal tracking in patients treated for lower extremity musculoskeletal cancer. MMC moderately agreed with a clinically accepted standard assessment yet requires further work to overcome some technical limitations. Patients who responded with the feedback forms found MMC acceptable and comfortable but a larger study with a higher response rate is required to confirm these findings. Some of the MMC measures obtained in this study such as unilateral stance balance and temporal measures were able to discriminate between major tumour groups suggesting early indicators of discriminant validity. Furthermore, some MMC variables of temporal, balance, gait and movement velocity showed significant relationships with established sarcoma scales suggesting early indicators of clinical convergent validity. These findings

and further validation work need to be undertaken in a future sufficiently powered study. This can further extend the use of MMC for clinical translation in busy clinics and in the community, to enhance rehabilitation practices in the sarcoma community.

## Supporting information

**S1 Checklist. STROBE statement—checklist of items that should be included in reports of observational studies.**
(DOCX)

**S1 Fig.** A: A-P view of anatomical landmark detection using MMC. B: M-L view of anatomical landmark detection using MMC.
(TIF)

**S2 Fig.** A: Raw MMC Data and Processed outcomes during walks of TUG test. B: Raw MMC Data and Processed outcomes during Unilateral Stance test. C: Raw MMC Data and Processed outcomes during Stand to Kneel and Kneel to Stand test.
(TIF)

**S1 Table. Skeletal tracking data.**
(PDF)

## Acknowledgments

Thanks to the staff of the North of England Bone and Soft Tissue Tumour Service for their assistance in patient screening and recruitment, in clinics, staff from the Orthopaedic Research Unit for their support in data collection and staff from Clinical Ageing Research Unit in Newcastle University for their technical and engineering input during data processing and analysis. Thanks to the University College London and Royal National Orthopaedic Hospital NHS Trust for support with resources for data analysis and write-up of this valuable piece of work.

## Author Contributions

**Conceptualization:** Sherron Furtado, Brook Galna, Lynn Rochester, Craig Gerrand.

**Data curation:** Sherron Furtado, Craig Gerrand.

**Formal analysis:** Sherron Furtado, Brook Galna, Lynn Rochester, Craig Gerrand.

**Funding acquisition:** Sherron Furtado, Lynn Rochester, Craig Gerrand.

**Investigation:** Sherron Furtado, Brook Galna, Lynn Rochester, Craig Gerrand.

**Methodology:** Sherron Furtado, Brook Galna, Alan Godfrey, Lynn Rochester, Craig Gerrand.

**Project administration:** Sherron Furtado, Craig Gerrand.

**Resources:** Sherron Furtado, Craig Gerrand.

**Software:** Sherron Furtado, Brook Galna.

**Supervision:** Sherron Furtado, Alan Godfrey, Lynn Rochester, Craig Gerrand.

**Validation:** Sherron Furtado, Brook Galna, Lynn Rochester, Craig Gerrand.

**Visualization:** Sherron Furtado, Brook Galna, Craig Gerrand.

**Writing – original draft:** Sherron Furtado.

**Writing – review & editing:** Sherron Furtado, Brook Galna, Alan Godfrey, Lynn Rochester, Craig Gerrand.

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
