## [Decision Letter · Decision Letter 0]

25 Jul 2023

PONE-D-23-14657Feasibility of Using Low-cost Markerless Motion Capture for Assessing Functional Outcomes after Lower Extremity Musculoskeletal Cancer SurgeryPLOS ONE

Dear Dr. Furtado,

Thank you for submitting your manuscript to PLOS ONE. After careful consideration, we feel that it has merit but does not fully meet PLOS ONE’s publication criteria as it currently stands. Therefore, we invite you to submit a revised version of the manuscript that addresses the points raised during the review process.

We look forward to receiving your revised manuscript.

Kind regards,

Roberto Di Marco, PhD

Academic Editor

PLOS ONE

Journal Requirements:

"The institution of one or more of the authors (AG, LR) has received, during the study period, funding from the National Institute for Health Research (NIHR) Newcastle Biomedical Research Centre and Unit based at Newcastle upon Tyne Hospitals NHS Foundation Trust and

Newcastle University. The institution of one or more of the authors (AG, SDD, LR) has received, during the study period, funding from the NIHR Newcastle Clinical Research Faculty infrastructure funding. The institution of one or more of the authors (CG, SF) has received, during the study period, funding from the Children with Cancer UK Charity, Sarcoma UK Charity, and Shear’s Foundation and Research and Capability (RCF).

All ICMJE Conflict of Interest Forms for authors are on file with the publication and can be viewed on request.

Each author certifies that his or her institution approved the human protocol for this investigation and that all investigations were conducted in conformity with ethical principles of research.

This work was performed at the major clinic sites and Human Movement Room at the North of England Bone and Soft Tissue Tumor Service, Newcastle Upon Tyne

Hospitals NHS Foundation Trust, UK."

Please confirm that this does not alter your adherence to all PLOS ONE policies on sharing data and materials, by including the following statement: ""This does not alter our adherence to  PLOS ONE policies on sharing data and materials.” (as detailed online in our guide for authors http://journals.plos.org/plosone/s/competing-interests).  If there are restrictions on sharing of data and/or materials, please state these. 

Please note that we cannot proceed with consideration of your article until this information has been declared. 

4. We noted in your submission details that a portion of your manuscript may have been presented or published elsewhere:

"This Paper relates to two published studies on accelerometery as analyses different datasets from the same study sample, but is different from those published as this paper investigates a different technology and approach MMC. I have included the two related papers in this submission."

Please clarify whether this publication was peer-reviewed and formally published. If this work was previously peer-reviewed and published, in the cover letter please provide the reason that this work does not constitute dual publication and should be included in the current manuscript.

6. We note that Figure S1 Fig 1A in your submission contain copyrighted images. All PLOS content is published under the Creative Commons Attribution License (CC BY 4.0), which means that the manuscript, images, and Supporting Information files will be freely available online, and any third party is permitted to access, download, copy, distribute, and use these materials in any way, even commercially, with proper attribution. For more information, see our copyright guidelines: http://journals.plos.org/plosone/s/licenses-and-copyright.

(1) You may seek permission from the original copyright holder of Figure S1 Fig 1A to publish the content specifically under the CC BY 4.0 license. 

(2) If you are unable to obtain permission from the original copyright holder to publish these figures under the CC BY 4.0 license or if the copyright holder’s requirements are incompatible with the CC BY 4.0 license, please either i) remove the figure or ii) supply a replacement figure that complies with the CC BY 4.0 license. Please check copyright information on all replacement figures and update the figure caption with source information. 

If applicable, please specify in the figure caption text when a figure is similar but not identical to the original image and is therefore for illustrative purposes only.

**Additional Editor Comments:**

Specifically, Reviewers are asking more clarity and details at some stages of the manuscript. The manuscript and the usability of the propose system would also benefit from a clear definition of the equipment performances (accuracy and precision of MMC system -as taken from literature would be acceptable-, influence of depth sensor on keypoints tracking, robustness to presence of third subjects, ...).

Reviewers' comments:

Reviewer's Responses to Questions

**Comments to the Author**

1. Is the manuscript technically sound, and do the data support the conclusions?

Reviewer #1: No

Reviewer #2: Partly

Reviewer #3: Yes

2. Has the statistical analysis been performed appropriately and rigorously? 

Reviewer #1: I Don't Know

Reviewer #2: Yes

Reviewer #3: Yes

3. Have the authors made all data underlying the findings in their manuscript fully available?

Reviewer #1: Yes

Reviewer #2: Yes

Reviewer #3: Yes

4. Is the manuscript presented in an intelligible fashion and written in standard English?

Reviewer #1: No

Reviewer #2: Yes

Reviewer #3: Yes

5. Review Comments to the Author

Reviewer #1: The authors used Markerless Motion Capture (MCC) to assess functional outcomes after sarcoma surgery in a cross section of 34 patients. The authors conclude that walking, single leg stance and kneeling assessment was feasible, acceptable and ‘clinically useful’.

This is an interesting technology that could have potential to improve our understanding of functional limitations after sarcoma surgery.

However, the manuscript fails to clearly communicate and interpret the study data. In addition, the study population is extremely heterogeneous (soft-tissue sarcoma, bone sarcoma, amputation, limb-salvage, all included). A more focused patient population would significantly improve the value of this study with respect to clinical relevance.

Abstract

Line 73: (19-89) is this an age range, interquartile range? Standard deviation?

Line 75: please clarify ‘clinically useful data’.

What do the ‘*’s mean in the abstract?

Overall, the abstract fails to connect the results of the analyses with the conclusions. What is the connection between the results and ‘novel knowledge’, ‘overcoming limitations’ and ‘guiding clinical management and rehabilitation’?

Introduction

Line 123: Please clarify ‘inexpensive’. What costs specifically are reduced with the Microsoft Kinect?

Overall, the rationale for the study is well explained and well supported by background information.

Methods

Lines 138-139: Please clarify why a new paper is required from the data accessed, given the 2 previous publications.

The pilot sample was one of convenience. That would exclude any statistical analysis. Therefore, statements such as ‘measures sensitive enough to significantly differentiate between tumour groups’, need further rationale/clarification. Same comment for lines 185-186.

Lines 165-166: Please provide more information on the patient feedback survey.

Lines 220-223: Statistical Analysis. Please clarify study power if stats were performed, with a sample size that was one of convenience.

Results

Line 240: Please provide some information on the cohort that declined to participate.

Line 251: Only 19 of 34 patients completed the survey. Please comment on this relatively poor response rate.

Overall, the Results section is difficult to follow, with extensive technical jargon. If the data is meant to improve outcomes for orthopaedic oncology patients, the language should be appropriate for orthopaedic oncologists who treat them, not kinematic and kinetic specialists.

Discussion

Lines 311-318: the study conclusions are strong, given that the data is not well interpreted for the reader. In addition, the low response rate weakens the ‘acceptability’ conclusions.

Lines 401-425: This section, case studies, is the most useful for the clinician: how can this technology be used to improve patient outcomes?

Overall, the Discussion is 3-4 X longer than appropriate. It is unfocused and disjointed. I suggest a complete rewrite.

Conclusions

The conclusions are difficult to assess, given the lack of clear interpretation of study results.

Reviewer #2: I congratulate authors on this new and promising tool that could help in monitoring effectivity of targeted physical therapy for sarcoma patients. These are my comments to authors :

1. The study was performed only on patients who had a lower limb surgery. Could the authors elaborate if any ongoing studies involve the evaluation of patients following surgery for upper limb sarcomas or not and do they think MMC could be equally effective in this group of patients.

2. Although sample size was justified based on the study being a pilot study (looking at feasibility, safety and patient compliance), statistical conclusions regarding convergent and discriminant validities based on this sample size may lack sufficient power.

3. I suggest that in future continuing research, using healthy subjects as controls would be recommended.

4. Seven patients were included in amputation group. Could you confirm if all 7 patients had the same type of amputation or not. An above knee amputation may have a functional outcome different from a below knee amputation.

5. Although your study showed statistical significance in discriminating between Bone and soft tissue sarcomas as a matter of functional outcome, from clinical point of view , the difference in outcome would rely more on the extent of surgery and reconstruction rather than the primary diagnoses of the disease. The functional outcome may be the same in large sized sarcomas whether primarily affecting bone or soft tissues.

Reviewer #3: The manuscript presents an interesting markerless motion capture method to assess functional outcomes after sarcoma surgery. The work highlighted the feasibility of using markerless motion capture based on Kinect technology to assess impairments in patient movements. The study is relevant to be published in the journal.

However, in my opinion, I propose some suggestions that could improve this work.

It will be helpful to the reader if you could briefly describe the established sarcoma assessments TESS and MSTS (lines 91-92). I suggest to slightly elaborate the Introduction section (lines 117-121) to what concern the review of the current literature that use the Microsoft Kinect in movement analysis, adding also relevant recent works, e.g. from the last five years.

The overall presentation of results is clear. The Authors also identify the limitations of their study. I suggest to modify at line 517-518 “current covid 19 pandemic” with “pandemic conditions” in order to keep it more general.

6. PLOS authors have the option to publish the peer review history of their article (what does this mean?). If published, this will include your full peer review and any attached files.

Reviewer #1: No

Reviewer #2: **Yes: **ahmed mohamed el ghoneimy

Reviewer #3: No

---

## [Author Response · Author response to Decision Letter 0]

23 Jan 2024

Dear Reviewer and Editor.

Thank you to you all for reviewing my manuscript. I have now responded to all reviewers queries and also undertaken the Major Revisions on this Manuscript as requested.

Please see attached a 'Response to Reviewers Document' (at the end of the PDF) we have compiled to address the specific comments made by you all Reviewers and Editors. 

This document contains responses to your comments, along with signposting you to the Major Revisions undertaken on the Manuscript and attachments - as per your request.

Please do not hesitate to contact me if any queries.

Many thanks.

Kind regards

Sherron Furtado

---

## [Decision Letter · Decision Letter 1]

27 Feb 2024

Feasibility of Using Low-cost Markerless Motion Capture for Assessing Functional Outcomes after Lower Extremity Musculoskeletal Cancer Surgery

PONE-D-23-14657R1

Dear Dr. Furtado,

We’re pleased to inform you that your manuscript has been judged scientifically suitable for publication and will be formally accepted for publication once it meets all outstanding technical requirements.

Kind regards,

Roberto Di Marco, PhD

Academic Editor

PLOS ONE

Additional Editor Comments (optional):

Reviewers' comments:

Reviewer's Responses to Questions

**Comments to the Author**

1. If the authors have adequately addressed your comments raised in a previous round of review and you feel that this manuscript is now acceptable for publication, you may indicate that here to bypass the “Comments to the Author” section, enter your conflict of interest statement in the “Confidential to Editor” section, and submit your "Accept" recommendation.

Reviewer #1: All comments have been addressed

Reviewer #2: All comments have been addressed

Reviewer #3: All comments have been addressed

2. Is the manuscript technically sound, and do the data support the conclusions?

Reviewer #1: Yes

Reviewer #2: Yes

Reviewer #3: (No Response)

3. Has the statistical analysis been performed appropriately and rigorously? 

Reviewer #1: Yes

Reviewer #2: Yes

Reviewer #3: (No Response)

4. Have the authors made all data underlying the findings in their manuscript fully available?

Reviewer #1: Yes

Reviewer #2: Yes

Reviewer #3: (No Response)

5. Is the manuscript presented in an intelligible fashion and written in standard English?

Reviewer #1: Yes

Reviewer #2: Yes

Reviewer #3: (No Response)

6. Review Comments to the Author

Reviewer #1: Thank you for addressing my comments. The paper is clearer with a more clinically relevant message and the data is well tied to conclusions.

Reviewer #2: Thank you for addressing most of the comments in revised manuscript . It can be premature or not clear how the outcomes of MMC can be scored or translated into recommendations that can be useful for monitoring post-operative rehabilitation programs and guiding patients and surgeons into expectations following different types of surgeries, but will be looking forward how the authors will address these issues in their future studies.

Reviewer #3: (No Response)

7. PLOS authors have the option to publish the peer review history of their article (what does this mean?). If published, this will include your full peer review and any attached files.

Reviewer #1: **Yes: **Michelle Ghert

Reviewer #2: **Yes: **ahmed mohamed el ghoneimy

Reviewer #3: No

---

## [Editor Report · Acceptance letter]

19 Mar 2024

PONE-D-23-14657R1 

PLOS ONE

Dear Dr. Furtado, 

I'm pleased to inform you that your manuscript has been deemed suitable for publication in PLOS ONE. Congratulations! Your manuscript is now being handed over to our production team.

Kind regards, 

on behalf of

Dr. Roberto Di Marco 

Academic Editor

PLOS ONE